# Lime-Based Mortar Reinforced by Randomly Oriented Short Fibers for the Retrofitting of the Historical Masonry Structure

**DOI:** 10.3390/ma13163462

**Published:** 2020-08-06

**Authors:** Michele Angiolilli, Amedeo Gregori, Marco Vailati

**Affiliations:** Department of Civil, Building and Environmental Engineering, University of L’Aquila, 67100 L’Aquila, Italy; michele.angiolilli@graduate.univaq.it (M.A.); marco.vailati@univaq.it (M.V.)

**Keywords:** cultural heritage, durability, mechanical characterization, retrofitting, strengthening, quasi-brittle material, three-point bending test, energy fracture, NHL, composite material

## Abstract

Recent seismic events prompted research to develop innovative materials for strengthening and repair of both modern and historic masonry constructions (buildings, bridges, towers) and structural components (walls, arches and vaults, pillars, and columns). Strengthening solutions based on composite materials, such as the Fiber Reinforced Polymers (FRP) or the Fiber Reinforced Cementitious Matrix (FRCM), have been increasingly considered in the last two decades. Despite reinforcement made of short-fibers being a topic that has been studied for several years from different researchers, it is not yet fully considered for the restoration of the masonry construction. This work aims to experimentally investigate the enhancement of the mechanical properties of lime-based mortar reinforced by introducing short glass fibers in the mortar matrix with several contents and aspect ratios. Beams with dimensions of 160 mm × 40 mm × 40 mm with a central notch were tested in three-point bending configuration aiming to evaluate both the flexural strength and energy fracture of the composite material. Then, the end pieces of the broken beams were tested in Brazilian and compressive tests. All the tests were performed by a hydraulic displacement-controlled testing machine. Results highlight that the new composite material ensures excellent ductility capacity and it can be considered a promising alternative to the classic fiber-reinforcing systems.

## 1. Introduction

The use of masonry is very common in many historic constructions, both in architectural monuments and whole urban centers all over the world. This masonry is generally made of various and very poor materials, characterized by different typologies. The fragility of this heterogeneous material interferes with the ductility criteria based on energy dissipation, which nowadays constitutes the safety principles of structural design for the safeguarding of human lives [1]. The disasters generated by seismic actions discouraged the use of unreinforced masonry from ancient times until the more modern era. Suffice it to say that the adoption of retrofitting systems began with the primordial civilization, such as traditional earthquake-resistant timber frames [2,3].

The development of a fiber-based strengthening system began in the 1960s when the potential for adding steel fibers to enhance the ductility of concrete material was recognized. However, this technology has been commonly adopted for the reinforcement of masonry structures only in the last decade as an alternative to traditional systems, such as mortar injections, reinforced drilling, and reinforced concrete plaster. Indeed, because of the strict rules for the preservation of historic structures, conservation committees usually request structurally efficient but less intrusive techniques to protect the historical structures.

Among modern and innovative solutions of intervention on existing structures, composite materials, such as the Fiber Reinforced Polymers (FRP) or the Fiber Reinforced Cementitious Matrix (FRCM), have been increasingly considered for strengthening and repair of both modern and historic masonry constructions (buildings, bridges, towers) and structural components (walls, arches and vaults, pillars, and columns). These technologies consist of the use of composites material characterized by uni- or bi-directional long fibers.

These materials are proven to be effective in increasing the load-carrying capacity of masonry elements and improving their structural behavior through a reduction of critical brittle failure modes. Most importantly, the increase in strength is obtained with a lower increment of the structural weight, as compared to the traditional ones (e.g., reinforced concrete plaster).

The FRP technology consists of the application of laminates and rods. On one hand, the use of laminates involves the application of fiber sheets by manual lay-up to the surface of the masonry panels, which is previously prepared by sandblasting and puttying procedure. The fibers are impregnated by an epoxy resin, which after hardening enables the newly formed laminate to become an integral part of the strengthened member. On the other hand, the use of pultruded rods consists of placing them into grooves cut onto the surface of the member being strengthened. The groove is filled with an epoxy-based paste, and the rod is then placed into the groove and lightly pressed to force the paste to flow around the rod. The groove is then filled with more paste and the surface is leveled [4].

High tensile strength and stiffness-to-weight ratio, fatigue and corrosion resistance, easy in-situ feasibility and adaptability, and progressive reduction in production and distribution costs are the main characteristics that encouraged the diffusion of the FRPs [5]. The FRP is employed to improve the global behavior in the seismic zone (tying, connections among components, strengthening), to counteract specific incipient or developed damage (high compression, shear, and/or flexural conditions), and to repair very specific local weaknesses depending on the peculiar construction typology [4,6,7,8].

However, the FRPs’ low fire resistance, high sensitivity to ultraviolet radiation when exposed to the open air, high toxicity, low vapor diffusion, and relatively short shelf life constitute obvious disadvantages for this retrofitting system [9]. As the executive phase is concerned, it is worth noting that the laying of the FRP materials must take place on completely dry surfaces, preventing the epoxy resin from coming into contact with moist parts; otherwise, the adhesion may be compromised.

Most of the drawbacks listed above are mainly related to the epoxy matrix used to embed and bond the fibers. That material is completely not compatible with the chemical property of the ancient mortars, leading to a severe breathable issue of the masonry walls. Thus, substituting the epoxy matrix with a mortar matrix appeared to be the most reasonable solution to improve the overall performance of externally-bonded composite systems. Furthermore, due to the nature of the FRP installation, the fracture may be caused in some areas where the masonry wall is not strengthened, particularly in the case of very brittle masonries, such as the irregular stone masonry.

Presently, the application of Fiber Reinforced Cementitious Matrix (FRCM) may overcome the disadvantages observed for the FRPs and represents the most favored choices in many projects [10]. Indeed, recent research [9,10,11,12,13,14,15,16,17,18,19,20] revealed the mechanical efficiency of the FRCM, its resistance to high temperatures and radiation, high vapor diffusion ability, and the possibility to perform installation even on a wet substratum.

In the FRCMs, the long fibers are embedded in a mortar matrix capable of ensuring the adhesion with the support. The function of the fibers is to carry tensile stresses, whereas the function of the matrix is to encapsulate and protect the fibers and transfer stresses from the mortar or masonry substrate to the fibers.

In FRCM composite systems, the fiber sheets or fabrics that are typically used in FRP are replaced with open fabric meshes in which the strands are spaced in both vertical and horizontal directions forming a bidirectional orthogonal grid. The behavior of masonry walls reinforced by FRCM tested under diagonal compression can be differentiated in three phases: (i) the load is carried mainly by the mortar matrix until cracking; (ii) the matrix undergoes a multi-cracking process resulting in the transfer of stresses from the matrix to the fibers; and (iii) the load is carried almost exclusively by the fabric [20].

Despite all advantages that this strengthening system can provide, it is characterized by a long application procedure that consists of three phases: (i) application of the first layer of mortar on the panel surfaces; (ii) application of the fiber grids on the panel surface lightly pressing them on the fresh mortar layer to have the fresh mortar passing through the grid openings; and (iii) application of a second finishing layer of mortar on the panel surfaces to cover the glass fabric while the previous mortar layer was still fresh. Even if this procedure can be considered easier than the ones concerned for the FRPs and concrete plaster, it still represents a limitation.

A common disadvantage of both the FRCMs and FRPs concerns the orientation of the fibers in specific directions: the FRCMs are characterized by fiber strands oriented in a bidirectional way; the FRPs are characterized by a prior defined fiber direction (usually along the diagonal and the edges of the wall). When the stress state is known, the proper use of such composites is expected to suitably orient the fibers in the direction of the maximum stress to optimize the efficiency of the material. Fibers activate their characteristics along their axial direction, whereas they have negligible properties in the other directions [5]. Hence, the composites with long fibers, such as the FRCM or FRP, are characterized by high resistance only in the direction of the fibers. However, stress may vary substantially in different load conditions. In particular, tensional states induced by seismic events do not act in a single and defined direction. In this case, the classic fiber-based systems may not be really efficient.

Therefore, it is necessary to consider the adoption of a diffuse reinforcement consisting of short fibers randomly oriented in the mortar matrix (discontinuous-fiber-reinforced composites) to ensure proper seismic capacity. Short and randomly distributed fibers can overcome the concern related to the material brittleness and poor resistance to crack initiation and growth [21].

The mechanical characterization of the lime-based mortar reinforced by randomly oriented short fibers is presented in this work, aiming to investigate the fiber content and the fiber type (different Aspect Ratio) on the flexural, tensile, and compressive strength as well as the energy fracture. Beams with dimensions of 160 mm × 40 mm × 40 mm with a central notch were prepared at the laboratory of the “Aquilaprem S.r.l.” company (L’Aquila, Italy). Then, the specimens were tested at the LPMS (Laboratorio di Prove Materiali e Strutture) of the University of L’Aquila. First, the samples were tested in three-point bending configuration. Then, the end pieces of the broken beams with a size of 80 mm × 40 mm × 40 mm were tested in compression and Brazilian configurations. Such an experimental procedure was also employed in [22,23]. A final comparison between all the mechanical properties is proposed and analyzed.

## 2. Description of the Newly Short-Fibers-Based-Strengthening-System

In recent years, considerable interest has been aroused by different nature of short-fibers (steel, plastic, glass, cast iron, polypropylene, polyacrylonitrile, polyolefin, etc.) to enhance the mechanical properties of cementitious materials, characterized by brittle nature with a low tensile strength and strain capacity [24]. In particular, the use of the fibers is greatly increased especially for concrete structures (e.g., industrial concrete slabs, structural or nonstructural precast elements and tunnel coatings).

Incorporation of fibers into cementitious materials can produce materials with increased modulus, increased strength for high fiber content, decreased elongation at rupture, increased hardness even with relatively low fiber content, and improvements in cut, tear, puncture, and impact load resistance [25]. The enhancement is mainly ensured by preventing or controlling the initiation and propagation of cracks [26]. Another advantage concerns the easier execution of structural elements, as compared to the traditional technology (based on the use of reinforcing bars and/or welded mesh).

The performance of a fiber-reinforced material, although in part related to the elastic properties of the fibers (depending on their nature), depends on many factors, such as fiber geometry, fiber content, fiber dispersion, fiber orientation, and fiber–matrix adhesion. Among the several factors, the bond behavior at the fiber–matrix interface plays a role of primary importance. Indeed, the ultimate elongation of the fibers is about 2–3 orders of magnitude higher than the ultimate strain of the mortar matrix and, therefore, the failure of the mortar matrix takes place before fiber failure. In particular, the adherence property of the mortar usually increases for high mechanical properties of the mortar, namely compressive and tensile strengths.

The fibers provide the greatest benefits especially in the softening phase when the maximum resistance of the material is achieved. In that phase, fibers are arranged astride the damage allowing the transmission of forces through a “sewing effect” that prevents the brittle collapse of the material (as one would observe in the absence of fiber reinforcement). Hence, the aspect ratio of the fibers, as well as their shape (fibers with bent ends, hooked and wavy fibers, etc.), assumes considerable importance in the load-bearing capacity, when the cracks through the material occur, affecting the anchoring of the fiber from the matrix and, consequently, yield more efficient the effect of the fiber on the mechanical behavior of the fiber-reinforced material.

The sewing effect also depends on the number of fibers that are arranged astride the damage. Therefore, both the fiber content and the fiber distribution in the cement play an important role in the mechanical behavior of the composite material. Obviously, the higher the fiber content, the higher the fiber distribution in the cement. Therefore, the higher the fiber distribution in the cement, the higher the efficiency of the fiber reinforcement. However, it worth noting that high quantities of fibers also produce a reduction in the fluidity of the fresh product. This aspect should be taken into account in the mix-design phase.

Despite reinforcement made of short-fibers being a topic that is being studied for several years from different researchers, no commercial product made of lime-mortar reinforced with short-fibers is nowadays employed for the strengthening of the existing historical structures.

The idea of using short-fibers embedded in a lime-based mortar matrix for the strengthening of the walls of the historical stone masonries can be considered as a promising newly reinforcement system that may ensure an adequate safety level for seismic forces acting in any direction. The new material was conceived to be compatible with the old constituent material of existing historical masonry. Indeed, the low compatibility of the cement-based mortar of the classic strengthening system with the lime-based mortar of the masonry joints yet represents an issue. In several recent cases, extensive damage occurred to the ancient masonry due to the incompatibility of the cement-based mortars [27,28,29]. Current standards [30,31] define cement-based mortars to be inadequate for strengthening interventions of historical masonries. Natural Hydraulic Lime (NHL)-based mortars [29,32,33] are considered a promising alternative to cement materials when high chemical-physical compatibility with historical substrates is strictly required.

The new composite material presents high flexibility in its application methodologies to the historical masonry structures. Indeed, it can be used as a coating to the masonry surfaces or in the structural repointing technique. The latter consists of replacing the deteriorated mortar or filling the missing mortar in the joints by employing the new composite material, allowing both to enhance the shear capacity of walls and preserve the original aesthetic of the masonry texture. Indeed, when choosing a retrofit method, its impact on the aesthetics of the building being retrofitted needs to be evaluated [4]. Aesthetic considerations are fundamental for historic structures. Many unreinforced masonry buildings are part of the cultural heritage of a determined city or country. Thereby, the preservation of their aesthetic and architecture is of main importance and retrofit work should be carried out with the least possible irrevocable alteration to the building’s appearance. It is recognized that the use of external reinforcing, such as the FRP or the FRCM, can alter the aesthetic of a masonry wall and resulted as unsatisfactory to retrofit churches and historical buildings after the last earthquakes in Europe. The use of the structural repointing by using the SFRLM is an alternative to strengthen masonry walls where aesthetics is an important issue.

## 3. Description of the Materials

Here, a description of the experimental investigation performed on the novel composite material for the retrofitting of masonry structures is presented. That material consisted of a lime-based mortar reinforced with short-fiber randomly oriented in the mortar matrix. In this study, zirconia-alkali-resistant glass fibers were employed. In particular, the effect of the aspect ratio of the fibers (i.e., the ratio between the length and the average diameter of the fibers) and the fiber content on both the tensile and compressive strengths as well as the workability of the product were investigated.

For the development of the product, the glass fibers were closely selected based on their length $\ell_{f}$. From the literature, it is known that the higher the fiber length, the higher the mechanical properties of the fibrous-product. However, the long fiber creates a problem in the mixing phase, especially with the goal to spray the fresh fibrous product as a coating to the masonry surface (the main possible application of this product), with a consequent loss in the application easiness or the total inapplicability of the product. Hence, the limit of the fiber length was an important factor considered in the development of the new material.

The diameters $d_{f}$ of the single yarn of both the glass fiber type were almost the same (ranging from 0.0135 to 0.014 mm). However, the fibers were originally impregnated with the matrix resin from manufactures. This has resulted in greater effective diameters of the glass fibers (about from 0.3 mm to 0.5 mm), whose values were not provided by manufacturers. Hence, a measurement of the effective diameters $d_{f}^{*}$ of the glass fiber was performed by measuring their diameters by an electronic micrometer. Actually, the cross-section of the fibers was not perfectly circular and therefore $d_{f}^{*}$ represents the diameter of the fiber with an ideal circle cross-section of the fiber.

Table 1 summarizes the geometry (fiber length $\ell_{f}$, diameters of the single yarn $d_{f}$ and the diameters of the strand $d_{f}^{*}$) and the mechanical properties (density $\rho_{f}$, Young’s Module $E_{f}$, tensile strength $f_{t,f}$, ultimate strain $\varepsilon_{u,f}$ and moisture content MC) of the two glass short-fibers (F1 and F2) used in the experimental campaign (illustrated in Figure 1a,b).

The following results were obtained by assuming the same values of content of lime (NHL 3.5 content equal to 30% of the total weight of the binder; the remaining 70% was a Portland cementitious binder), water content (equal to 80% of the binder weight), sieve curve (sand content equal to 65% of the total weight of the product with size ranging from 0.1 mm to 1.2 mm), and fluidizer content (equal to 0.2% of the total weight of the product). The natural hydraulic lime mortar was assumed in that content because it is intrinsically characterized by a higher variability of its mechanical properties [34], as compared to the cement. Hence, to better investigate the effect of the fiber type (the F1 and the F2) and the fiber content *F* on the mechanical properties of the composite material, authors decided to assume that lime content aimed to have results characterized by a lower dispersion of the data. Anyhow, that content was enough to obtain the so-called “lime-based” mortar instead of the “pure-lime mortar”.

Furthermore, the choice to employ the same mix-design for the reinforced and unreinforced mortar specimens was also due to the intention to better investigate the effect of the fiber properties. Therefore, fresh products were characterized by different consistency. In particular, *F* was assumed equal to 1.5%, 2.0%, and 2.5% of the total weight of the product (corresponding almost to 1%, 1.3%, and 1.6% of the total volume of the product, respectively) It is worth noting that, even for the unreinforced samples, the mix design of the product (water content, sieve curve, and content of additives) was the same as the fiber-reinforced ones. Hence, the enhancement of the mechanical properties of the fibrous mortar, as compared to the unreinforced mortar, was merely due to the fiber type and fiber content.

For each batch of the product, immediately after the slump test (described in Section 4.1), three mortar samples measuring 160 mm in length, 40 mm in height, and 40 mm in thickness were cast according to the standard code EN 1015-11 [35]. Fibers were added during the mixing phase. The specimens were cast in molds and were kept moist for 48 h in the environmental chambers. Next, the samples were demolded and left in laboratory conditions (room temperature and ambient humidity of about 20° and 60%, respectively) for 26 days, for a total age of 28 days, before testing in a three points bending test (Section 4.2). After the 3PBTs, the end pieces of the broken beams were used to determine their tensile strength (Section 4.3) and compressive strength (Section 4.4).

In particular, a total of 21 mortar specimens were prepared, namely three unreinforced samples and 18 fibrous mortar samples (three samples for each of the three fiber contents adopted for both the two fiber types).

## 4. Methods

### 4.1. Characterization of the Consistency

The slump test is an empirical method that measures the workability of fresh mortar (or fresh concrete). More specifically, it measures the consistency of freshly made mortar in a specific batch of the product that can be subsequently used for the mechanical characterization tests, namely three points bending test, compressive test, and direct-indirect tensile test. It is a term that describes the state of fresh mortar. In particular, it is the relative mobility or ability of freshly mixed mortar to flow. It includes the entire range of fluidity from the driest to the wettest possible mixtures [36]. Consistency is a term very closely related to workability.

Practical evaluation of the consistency of the paste can be performed according to the procedure reported in EN 1015-3 [35] and ASTM C1437 [37]. In particular, the so-called slump test consists of measuring the mean diameter of the fresh mortar, previously cast in a specific steel mold, after 20 strokes of the flow table. The measure of the diameter value of the fresh mortar at the end of the test represents the “slump” of the product. The higher the slump value, the higher the workability of the fresh mortar.

### 4.2. Characterization of the Flexural Strength

To evaluate the enhancement of the innovative fibrous lime mortar material in terms of its flexural strength as well as its fracture energy, three points bending test (3PBT) was carried out on several specimens measuring 160 mm in length, 40 mm in height, and 40 mm in thickness. A 2 mm thick notch ($t_{n}$) was fabricated on the mortar samples (by using wet sawing) with a depth *a* of 6 mm, resulting with a notch to beam depth ratio *a*/*d* of 0.15.

The notches were fabricated on the mortar samples to minimize irreversible deformations outside the fracture zone, avoiding large parts with high stresses outside this zone. For that issue, Hillerborg [38] suggested a depth notch of 0.3–0.4 times the beam depth. However, in the present research, the *a*/*d* ratio was chosen equal to 0.15, since the higher values of that ratio would have led to a larger dispersion of the results because of the presence of the short-fiber.

Indeed, the lower *a*/*d* ratio leads to an increase in the number of fibers passing through the unnotched ligament (d−a). Hence, one can easily understand that, by increasing the number of fibers passing through the unnotched ligament, less scattering of the results can be obtained. Furthermore, the *a*/*d* suggested by Hillerborg referred to concrete material, in which the ratio between the size of the fracture plane and the size of the maximum aggregate is different from the mortar case. The final important reason in the choice of a *a*/*d* ratio equal to 0.15 was due to the nature of the material: lime-based mortar beams would have easily broken during handling in case of deeper notches.

The scheme of the notched beam used for the 3PBT is illustrated in Figure 2a. In particular, the nominal distance between the supports *L* was 100 mm, whereas both the width *d* and the thickness *b* were equal to 40 mm. The loading *P* was introduced at the midspan of the beam. The two rollers at the bottom allowed for free horizontal movement.

The three-point bending test was performed on notched beams to determine the maximal tensile stress $\sigma_{f}$ as well as the fracture energy $G_{f}$. Indeed, as described by a Bazant’s work [39], the fracture energy of quasi-brittle material is a basic material characteristic needed for a rational prediction of brittle failures of such structures.

Failure of quasi-brittle structures generally consists of numerous micro-cracks that might result in fracturing of the structures under loads. Thus, a micro-crack in quasi-brittle material may become a potential source of crack propagation leading to probable catastrophic failure. Definitely, the failure mechanism can be studied by quantifying the energy consumed in crack propagation and the formation of new crack surfaces.

In principle, the fracture energy as a material property should be a constant, and its value should be independent of the method of measurement, various test methods, specimen shapes, and sizes. However, these variables lead to very different results (e.g., [39,40,41,42]).

Despite the scientific interest in chaotic stone constructions, the test method for the determination of $G_{f}$ and even its precise definition has been a subject of intense debate among researchers because it has been found to vary with the size and shape of the test specimen and with the test method used. The commonly used method for measuring the fracture energy is the work-of-fracture method recommended by RILEM [43,44], in which the total energy $G_{f}$ is evaluated by dividing the total applied energy by the projected ligament area, as follows:(1)$$G_{f}=\frac{W_{0}+m\;g\;\delta_{0}}{(d-a)\;b}$$
where $W_{0}$ is the area of the complete load-deflection curve, *m* is the weight of the beam between the supports, calculated as the beam weight multiplied by l/*L*, *g* is the acceleration due to gravity, $\delta_{0}$ is the deformation at the final failure of the beam, *d* is the beam height, *b* is the beam width, and *a* is the notch depth. For stable test performance and to obtain reliable test data, the self-weight compensation [45] was used.

The flexural stress $\sigma_{f}$ was calculated by using the following equation:(2)$$\sigma_{f}=\frac{3\;P_{}\;L}{2\;b\;{\left(d-a\right)}^{2}}$$

For $P=P_{MAX}$, this equation gives the flexural strength.

The 3PBTs were performed in a displacement controlled hydraulic testing machine, by using the Zwick Roell test machine of the LPMS (Laboratorio di Prove Materiali e Strutture) of L’Aquila (Figure 2b). The specimens were loaded at a constant displacement rate of 0.5 mm/min. Both the force *P* and vertical mid-span displacement (or deflection) δ were directly recorded through the test machine.

### 4.3. Characterization of the Tensile Strength

The tensile strength of quasi-brittle material, such as mortar or concrete, can be determined from different types of tests, namely direct or indirect tensile tests. The direct test concerns the execution of direct pull tests, whereas the indirect tensile test concerns the execution of splitting tensile tests (also called the diametrical compression test, split-tension test, and Brazilian test (BT) among other names).

There are many technical difficulties in executing a true tensile strength test. A uniform stress distribution which makes it possible to calculate the true tensile strength is difficult to obtain. The method commonly used to determine tensile properties of quasi-brittle material is the flexural beam test by three-point loading on a beam over a span (the 3PBT described in Section 4.2). The flexural strength is computed from the bending moment at failure, assuming an ideal straight line stress distribution according to Hooke’s law. However, the calculated flexural strength may be higher than the true tensile strength [46].

Many attempts have been made to find a substitute for the 3PBT and the splitting tensile test of a cylindrical specimen may be the solution to the problem [47]. The splitting tensile strength test method has many merits compared with the direct tensile test method; for example, it can be conducted much more easily and the scatterings of the test results are very narrow. The BT is straightforward and economic and can be used on cylindrical specimens (fabricated in molds or extracted concrete cores) or flat disk-shaped specimens as well as cubes or prisms [48]. In addition, the test can be performed with the same machine that is used to perform direct compression tests, and samples identical in shape and geometry as those used in direct compression can be employed. The BT is useful to experiment brittle or quasi-brittle materials that have a much greater compression strength than their tensile strength and that are susceptible to brittle ruptures. Researchers have indicated that, among the three testing methods (direct tensile, splitting tensile, and flexural tests), the splitting tensile test gives the most accurate measurement of the true tensile strength of mortar or concrete materials in a wide strain rate [48].

In the BT, the sample is compressed with load concentrated on a pair of antipodal points. In this way, tensile stress is induced in the direction perpendicular to the applied load, and it is proportional to the magnitude of the applied load.

When the induced stress exceeds the tensile strength, fracture initiates at the geometric center of the sample. In agreement with the Griffith criterion [49], the exact center of the sample is the only point at which the conditions for failure under tension are satisfied because, in this site, the tensile stress equals the uniaxial strength of the tested material. Indeed, the BT result is accepted if fracture initiates at the center of the sample, and in this case, the measured value is representative of the tensile strength of the tested material. In the BT, the specimen must fail along the vertical line between compression points; otherwise, the observed failure mode is considered invalid. In the case of unreinforced specimens, the test typically ends with a sudden failure of the specimen when it reaches the maximum load due to the propagation of an unstable crack. Since its invention, the BT has motivated a wide variety of studies. One can gain an idea of its impact if one considers that the use of concrete test specimens has been standardized into norms in various countries, such as UNI EN 12390-6, ASTM C-496. However, the BT is far from a universal test, and it is unknown whether a geometric configuration exists that favors effective, robust testing that is less sensitive to other experimental parameters [50].

In the present research, to obtain the tensile strength of the mortar specimens, instead of performing a direct tension test, which is of needless difficulty, Brazilian Tests (BT) (ASTM C496) were conducted using 40 mm × 40 mm × 80 mm prismatic mortar specimens obtained from the two-half specimens tested in 3PBT (see Figure 3a,b).

The tensile stress $f_{t}$ was calculated by using the following equation:(3)$$f_{t}=\frac{2\;P_{}}{\pi \;b\;d}$$

By Equation (Equation 3), one can compute the tensile strength when $P=P_{max}$. The BTs were performed in a displacement controlled hydraulic testing machine, by using the Zwick Roell test machine of the LPMS of L’Aquila. The specimens were loaded at a constant displacement rate of 0.5 mm/min. Both the force *P* and vertical mid-span displacement (or deflection) δ were directly recorded through the test machine.

### 4.4. Characterization of the Compressive Strength

To have a complete overview of the mechanical behavior of the fibrous lime mortar material, the Compression Tests (CT) were also carried out to obtain the compressive strength of the specimens. In particular, the tests were performed on 40 mm × 40 mm × 80 mm prismatic mortar specimens obtained from the two-half-length specimens result from 3PBT after testing the full-length specimen (Figure 4).

The distributed load *P* was applied on a squared area of 40 mm, while the specimens were placed on a squared area of 40 mm. One can compute the compressive stress by using the following equation:(4)$$f_{c}=\frac{P_{}}{b\;L^{\prime \prime }}$$

By Equation (Equation 4), one can compute the compressive strength when $P=P_{max}$. Moreover, the vertical strain $\varepsilon_{v}$ is computed by the following equation:(5)$$\varepsilon_{v}=\frac{\delta }{d}$$

The CTs were performed in a displacement controlled hydraulic testing machine adopting a constant displacement rate of 1 mm/min. Even in this case, both *P* and δ were directly recorded by the test machine.

## 5. Results and Discussion

In Figure 5, one can observe the results of the slump tests carried out for the mortar specimens strengthened by the F1 and the F2 fibers. In that figure, the slump value is related to the fiber content *F* adopted in the mix-design of the fibrous mortar. It is worth noting that a linear variation of the slump (indicated by the dotted lines) was assumed from the case of the unreinforced mortar (*F* = 0%) to the reinforced mortar with the content fiber of 1.5% as no tests were carried out by using fiber content in that range (0%–1.5%).

As expected, in Figure 5, one can observe a decrease in the slump value by increasing the fiber content. This trend can be observed for both the fiber types. For the F1 case, one can see that no strong differences in terms of slump can be noted by comparing the unreinforced samples (*F* = 0%) and the *F* = 1.5% case. This means that a significant compaction effect of the F1 fiber on the mortar matrix can be achieved only with fiber content higher than 1.5%. This is obviously related to the content of additives introduced in the product to increase its workability. Indeed, for lower additive content, one would observe a higher reduction of the slump value even for *F* lower than 1.5%. Moreover, in Figure 5, one can instead observe an almost linear decrease in the slump value for the F2 case by increasing the fiber content.

Results plotted in Figure 5 indicate that the F1 fiber leads to higher workability of the product (higher slump value), as compared to the F2 one. This trend can be observed for all the fiber content *F*.

The large difference in the results plotted in Figure 5 between the two fiber types may be mainly due to their different aspect ratio AR. Indeed, for the same *F*, a higher value of the AR leads to a lower number of fibers per unit volume of the product, and vice-versa.

In particular, the volumes of the single fiber strands were 4.269 mm^3^ ($0.476^{2}\pi /4\times 24$) and 1.019 mm^3^ ($0.316^{2}\pi /4\times 13$) for the F1 and F2 fibers, respectively. It is worth noting that the density of the two fibers was the same (2680 kg/m^3^). Hence, at the same fiber content, the F2 fiber ensured almost four times the number of the fiber strands into the matrix of the mortar specimens, as compared to the F1. Therefore, due to the lower number of fibers, the F2 fiber allowed a higher compaction effect on the mortar matrix, as compared to the F1 fiber. This was reflected in the workability of the product: the lower the number of fibers, the higher the workability of the product.

Furthermore, the trend of Figure 5 may be influenced by the absorption level of the two types of fibers since they were produced by different manufacturers and different types of primers may have been employed for them. In particular, the moisture content (MC) declared from the two manufactures is equal to 0.6% and 0.5% for the F1 and F2 fibers, respectively, where MC is defined as the weight of water in a material express as a percentage of the total weight of the material (for more details on the effect of moisture content on the mechanical properties of glass fiber, the reader is referred to a recent study [51]). Hence, F1 fiber tends to absorb slightly more water than the F2 fiber. Despite this difference, the workability of the fresh mortar with F1 fiber is higher than the one with F2. Definitely, the higher workability obtained for the F1 fiber is mainly related to other factors, such as the number of the fibers into the mortar (depending on the fiber geometry).

Figure 6a shows the relation measured between the flexural stress $\sigma_{f}$ and the deflection δ obtained for the unreinforced mortar specimens under three-point bending tests (3PBT). Plots are referred to three tests. The mean value of the maximum flexural stress is equal to 2.9 MPa. Moreover, one can observe an almost perfect brittle behavior after the achievement of the maximum flexural stress. Indeed, the $\sigma_{f}$ suddenly drops after that point.

Figure 6b–d show the $\sigma_{f}$–δ plots obtained for the mortar specimens reinforced by the two fiber types (F1 and F2) tested under 3PBT by assuming different fiber content *F* (1.5%, 2.0%, and 2.5%). Plots are referred to a number of three tests for each fiber type and fiber content. One can see that, for F=1.5% and F=2.0%, no strong differences can be noted between the two different fiber types (F1 and F2) in terms of flexural strength. On the contrary, for the higher fiber content (F=2.5%), one can see that the F1 fiber leads to higher flexural strength, as compared to the F2 fiber. From those figures, it is clear that the F1 fiber leads to higher energy fracture for all the fiber content, as compared to the F2 fiber.

For a better interpretation of the effect of both the fiber content and fiber type on the mechanical properties of the composite material, one can see Figure 7a,b. In particular, Figure 7a shows the variation of the flexural strength $\sigma_{f}$, computed by using the equations Equation (Equation 2), as a function of the fiber content *F*. One can see that, for lower fiber content (F=1.5%), no differences can be noted between F1 and F2. Then, for higher values of fiber content (F=2.0% and F=2.5%), one can see different results by comparing the two fiber types. In particular, for the F1 case, one can observe a gradual increase in the flexural strength up to achieve almost 15 MPa (when F=2.5%). Instead, for the F2 case, a high increase in the flexural strength is observed from F=1.5% to F=2.0%, whereas one can observe a slight difference by comparing the case of F=2.0% and F=2.5%.

In general, from Figure 7a, one can see the benefit of the fiber content on the flexural strength albeit the variability of the results may be affected by many factors (i.e., specimens properties and failure mode).

Figure 7b shows the variation of the fracture energy $G_{f}$, computed by using the Equation (Equation 1), as a function of the fiber content. Respect to the results noted in terms of $\sigma_{f}$, one can observe a clearer trend in terms of $G_{f}$. For all the fiber content, one can see that the fracture energy computed for the F1 is more than two times the one computed for the F2. This is due to the higher contact surface of the F1 fiber with the mortar matrix. Indeed, the higher fiber length as well as the higher diameter of the F1 ensured a higher bond behavior (adherence level) at the fiber–mortar interface, as compared to the F2 fiber.

Figure 8a shows the relation measured between the tensile stress $f_{t}$ and the deflection δ obtained for the unreinforced mortar specimens under Brazilian Tests (BT). Plots are referred to two tests (one specimen was broken before testing). The mean value of the tensile strength is equal to 2.0 MPa. It is worth noting that the flexural strength computed in the 3PBT was equal to 2.9 MPa, which is 1.45 times higher than the tensile strength.

As already observed for the unreinforced specimens tested in 3PBT, one can observe an almost perfect brittle behavior. Indeed, $f_{t}$ suddenly drops after the achievement of the maximum tensile strength.

Figure 8b–d show the $f_{t}$–δ plots obtained for the fiber-reinforced mortar specimens tested under Brazilian Test (BT) by varying the fiber content therein the mortar matrix. Plots are referred to three tests (except the case of F=2.0% with the F1 fiber, for which two tests are presented).

For a better interpretation of the effect of both the fiber content and fiber type on the tensile behavior of the composite material, one can see Figure 9. In particular, that figure shows the variation of the tensile strength $f_{t}$, computed by using the Equation (Equation 3) for $P=P_{max}$, as a function of the fiber content.

One can see that higher values of tensile strength can be noted for the F2, as compared to the F1, for all the fiber content *F*. In particular, for the F2 case, one can observe a gradual increase in the tensile strength up to achieve almost 4 MPa for F=2.5%. It is worth noting that, for the same fiber content, a flexural strength equal to 11.4 MPa (2.85 times the $f_{t}$ value) was computed for the 3PBT. Instead, for the F1 case, one can observe no differences in the tensile strength between the F=1.5% and the F=2.0% cases, in which $f_{t}$ is equal to 2.5 MPa. Then, an increase in $f_{t}$ is observed for F=2.5%, in which $f_{t}$ is equal to 3.2 MPa. By comparing that value with the flexural strength computed for the same fiber content (F=2.5%), one can see a high difference ($\sigma_{f}$ is 4.8 times the $f_{t}$ value).

In general, from Figure 9, one can see the benefit of the fiber content on the tensile strength. It is worth noting that the F2 fiber type leads to the higher tensile strength though it was characterized by the lower fiber length and lower AR, as compared to the F1 fiber. This may be due to the number of fibers that the F2 fiber type ensured respect to the F1 one, at the same fiber content. Indeed, a higher number of fibers lead to higher compaction of the mortar matrix (as discussed for the results of Figure 5) as well as a higher number of the fiber over the entire projected ligament area, where cracks develop.

Figure 10a shows the relation measured between the compressive stress $f_{c}$ and the vertical strain $\varepsilon_{v}$ obtained for the unreinforced mortar specimens under Compression Test (CT). Plots are referred to four tests. Each test was interrupted at 40% of the maximum load in the post-peak behavior. One can compute a mean value of the compressive strength equal to 18.1 MPa. Moreover, one can observe a typical softening curve of quasi-brittle materials. Indeed, for all the curves, one can see that the compressive stress suddenly drops after the achievement of the maximum stress.

Figure 10b–d show the $f_{c}$–$\varepsilon_{v}$ plots obtained for the fiber-reinforced mortar specimens tested under Compression Test (CT) for different fiber content *F*. All the plots are referred to three tests. In general, from these results, one can see that the enhancement of the compressive strength due to both the fiber type and fiber content is more and more limited, as compared to the contribution offered by the fibers in the other mechanical properties. Furthermore, one can observe a large scattering of the results. In particular, an increase in the scattering of the results can be observed by increasing the fiber content. This trend was in line with what was expected. Indeed, the higher fiber content leads to lower homogeneity of the product with a consequent increase of the scatter in the results.

For a better interpretation of the effect of both the fiber content and fiber type on the compressive behavior of the composite material, one can see Figure 11. In particular, this figure shows the variation of the compressive strength $f_{c}$, computed by using Equation (Equation 4) for $P=P_{max}$, as a function of the fiber content.

As compared to the other results obtained for the 3PBT and the BT, the trends of the results obtained for the CT are less clear. Indeed, for the F1 case, one can see that, by increasing the fiber content from F=1.5% to F=2.0%, a reduction of the mean compressive strength is observed. The same unusual trend was also observed for the F2 case by increasing the fiber content from F=2.0% to F=2.5%. However, from a general point of view, one can see a clear increase in the compressive strength of the fiber-reinforced mortar as compared to the unreinforced case.

A final comparison of the results obtained by the three-point bending test (3PBT), Brazilian test (BT), and compression test (CT) is proposed in Figure 12a–d. In particular, this figure shows the increase (in percentage) in the flexural strength $\sigma_{f}$, fracture energy $G_{f}$, tensile strength $f_{t}$ and compressive strength $f_{c}$, as compared to the unreinforced case. One can see that the best benefit in introducing fiber therein the mortar matrix is observed in terms of the fracture energy (up to almost 2500% for the F1 case and 1000% for the F2 case). Excellent increase in the flexural strength can also be noted (up to almost 450% for the F1 case and 300% for the F2 case). As compared to the other mechanical properties, a lower increase in the tensile strength and especially in the compressive strength can be observed (less than 100% for all the cases).

It is worth noting that the variability of the results may be caused by many factors: specimen properties irregularity of the cross-section, the presence of micro-cracks, misalignment of the samples with respect to their mid-thickness, different thickness of the specimens, the randomness of the fibers, and last but not the least the spatial randomness of material properties. Hence, it was normal to observe a large dispersion of the results (that is obviously higher for the fibrous materials).

Definitively, despite the F2 fiber ensured, in most cases, the higher flexural, tensile, and compressive strengths, its energy fracture was clearly smaller, as compared to the F1 fiber. This result highlighted the importance to investigate the softening behavior of fibrous mortar specimens. Hence, the F1 fiber is more recommendable for the retrofitting system as compared to the F2 fiber.

Table 2 summarizes the mean and standard deviation values of the flexural strength, fracture energy, tensile strength, and compressive strength computed for the new composite material by assuming different fiber type (F1 and F2) and different fiber content (*F* = 0%, 1.5%, 2.0%, and 2.5%).

## 6. Conclusions

Fiber Reinforced Mortar has proven on more than on occasion that it can increase the strength, ductility, and toughness of plain mortar. This study was not presented just to prove this point. The ultimate goal was to understand how various types of fibers and their content affect the same mortar type and to consider these fibers in binders more closely related to material compositions of historical mortars, namely the lime-based mortar. In particular, three fiber contents were investigated: 1.5%, 2.0%, and 2.5% of the total weight of the product (corresponding almost to 1%, 1.3%, and 1.6% of the total volume of the product, respectively). Experimental results showed that the contribution of diffuse short fibers greatly increased (almost proportionally to their content) the mechanical properties of the lime-mortar. In particular, the fracture energy was the mechanical property that most benefited from the use of short fibers up to about 2500% (for fiber content equal to 2.5%), as compared to the unreinforced case. This value was computed for the fiber characterized by the longer length (24 mm). For the lower fiber length (13 mm), it was computed an energy fracture of about 2–3 times less. Then, for the highest fiber content, the maximum increase in the flexural and tensile strengths was about 450% and 100%, respectively. The mechanical property less affected by the fiber contribution was the compressive strength as its maximum increase was only about 50%.

It is worth noting that the long fiber strands used in the classic fiber-based strengthening system (namely, the FRCMs and the FRPs) have mainly the function of carrying tensile stresses. Hence, the results obtained for the new composite material are important as it can significantly increase both the strength and ductility of the mortar material. Definitely, the new material can ensure physical-chemical compatibility with the special characteristics of the historical masonries and also increasing their strength and ductility, by standing as a promising alternative to the classic fiber-reinforcing systems.

The overriding wish of the authors is to boost the use of sustainable materials and reduce the impact of the intervention in the strengthening of historical masonries. To do so, they will focus their next research on the mortars made with vegetable resins, natural yarns such as sisal, hempen, and similar, also using recycled materials.

## Figures and Tables

**Figure 1 materials-13-03462-f001:**
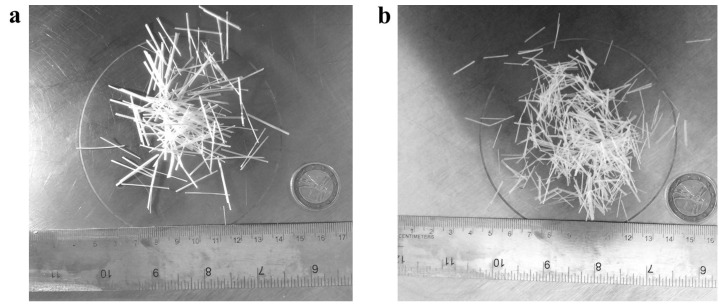
Two fiber types used in the experimental campaign: the F1 (**a**) and the F2 (**b**).

**Figure 2 materials-13-03462-f002:**
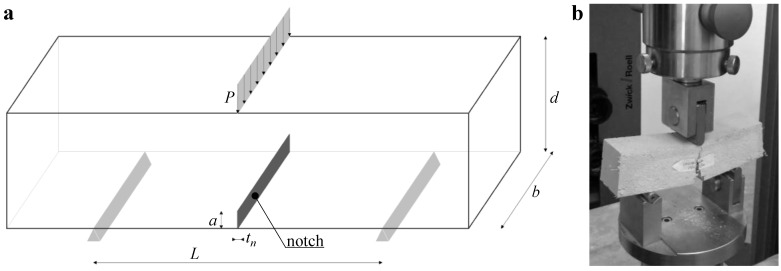
(**a**) notched beam adopted for the 3PBT; (**b**) picture of the 3PBT carried out at the LPMS of L’Aquila.

**Figure 3 materials-13-03462-f003:**
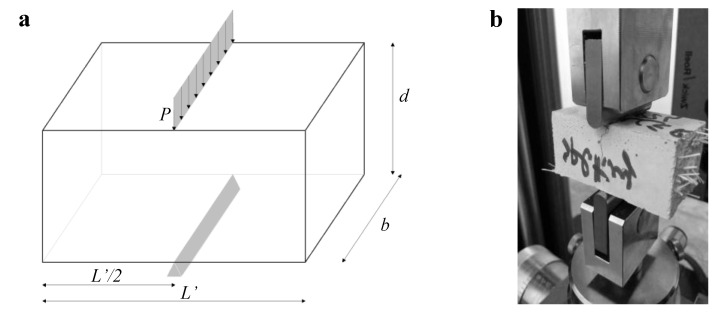
(**a**) scheme employed for the Brazilian test (BT); (**b**) picture of the BT carried out at the LPMS of L’Aquila.

**Figure 4 materials-13-03462-f004:**
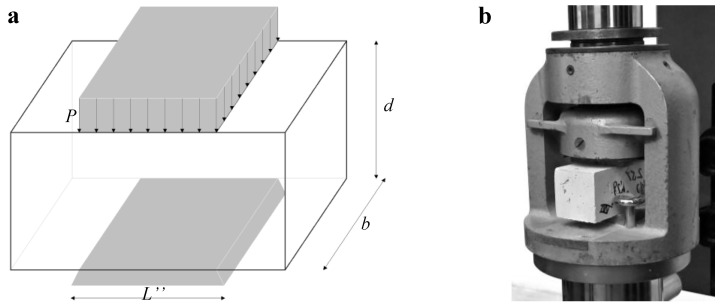
(**a**) scheme employed for the Compression Test (CT); (**b**) picture of the CT carried out at the LPMS of L’Aquila.

**Figure 5 materials-13-03462-f005:**
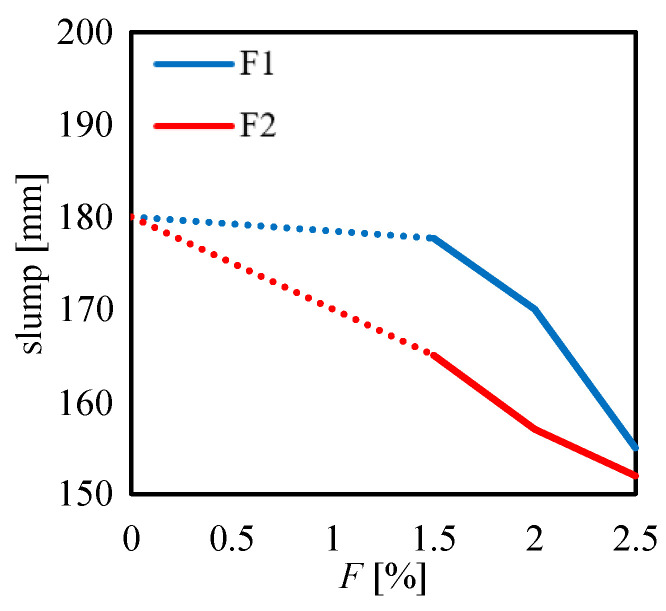
Slump measured for the F1 and the F2 fibers by varying the fiber content *F*.

**Figure 6 materials-13-03462-f006:**
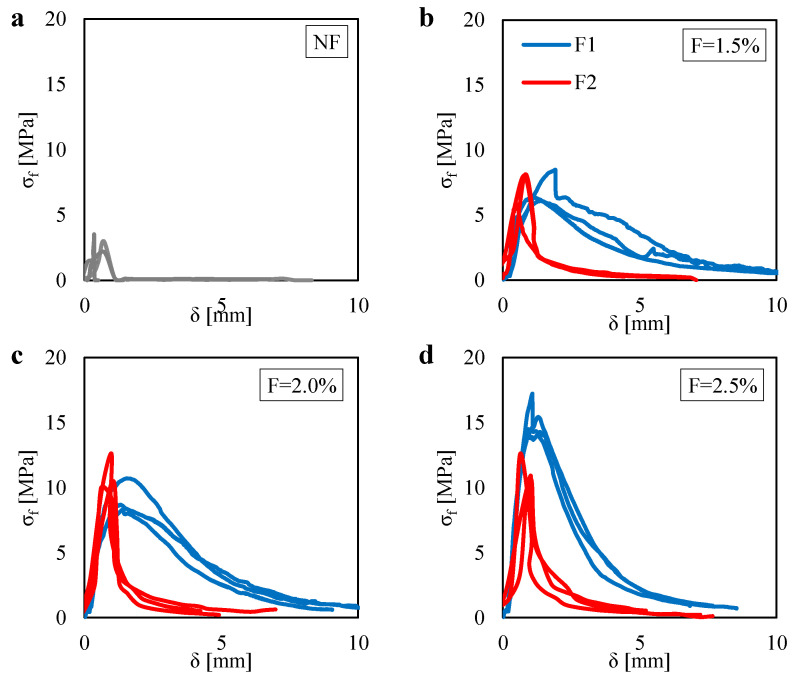
Flexural stress $\sigma_{f}$ related to the vertical displacement δ measured in the 3PBT for: (**a**) the unreinforced case (NF). Reinforced cases by assuming two fiber types (F1 and F2) with fiber contents equal to (**b**) 1.5%; (**c**) 2.0% and (**d**) 2.5%.

**Figure 7 materials-13-03462-f007:**
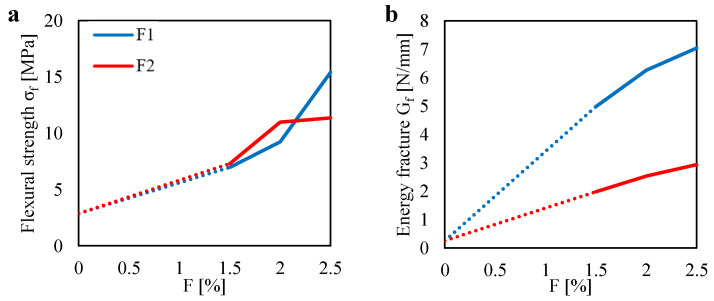
(**a**) flexural strength $\sigma_{f}$ and (**b**) fracture energy $G_{f}$ measured in the 3PBT for two fiber type (F1 and F2) with different fiber content.

**Figure 8 materials-13-03462-f008:**
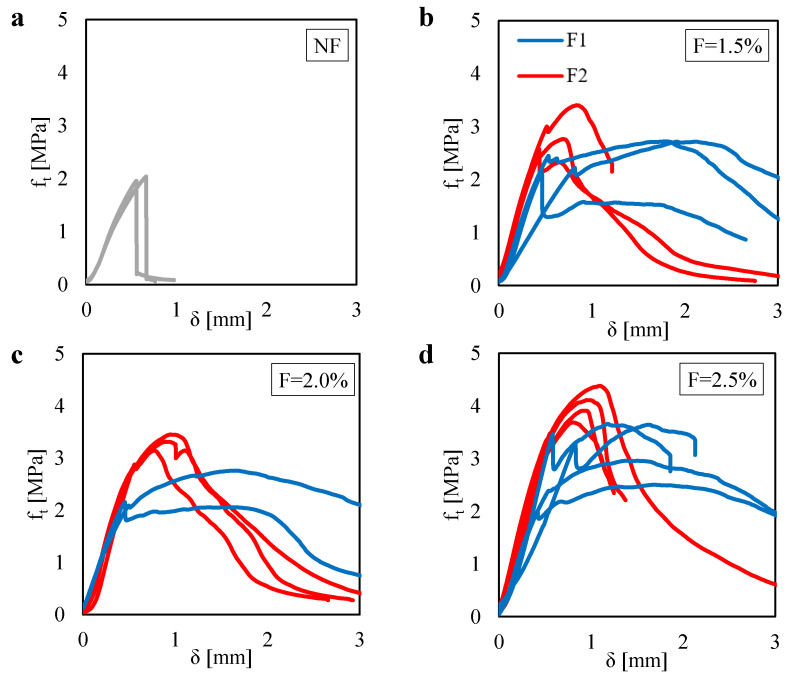
Tensile stress $f_{t}$ related to the vertical displacement δ measured in the BTs for: (**a**) the unreinforced case (NF). Reinforced cases by assuming two fiber types (F1 and F2) with fiber contents equal to (**b**) 1.5%; (**c**) 2.0% and (**d**) 2.5%.

**Figure 9 materials-13-03462-f009:**
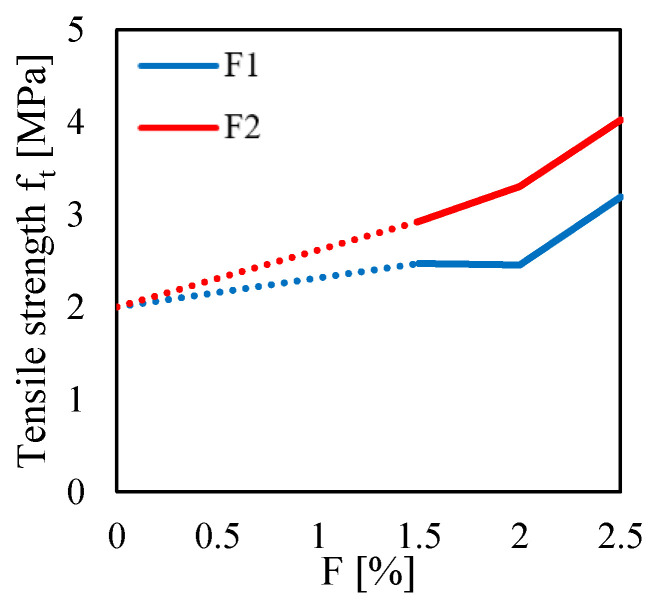
Tensile strength $f_{t}$ measured in the BTs for two fiber type (F1 and F2) with different fiber content.

**Figure 10 materials-13-03462-f010:**
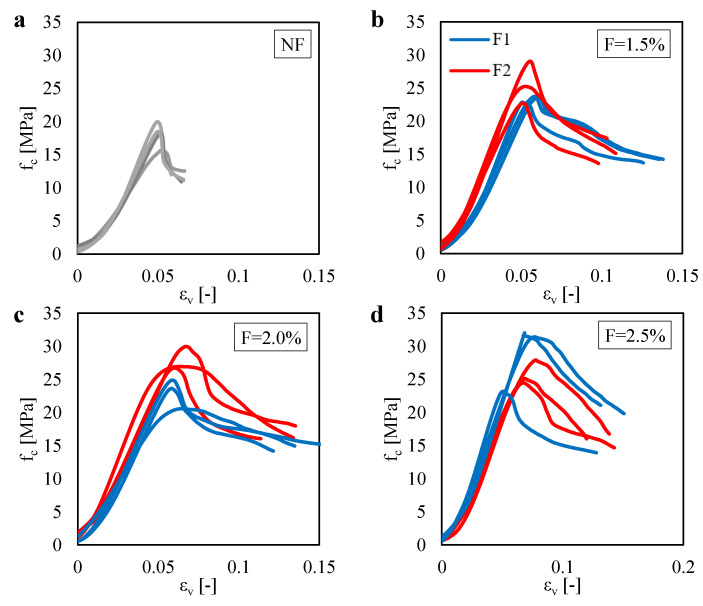
Compressive stress $f_{c}$ related to the vertical deformation $\varepsilon_{v}$ measured in the CTs for: (**a**) the unreinforced case (NF). Reinforced cases by assuming two fiber types (F1 and F2) with fiber content equal to (**b**) 1.5%; (**c**) 2.0%, and (**d**) 2.5%.

**Figure 11 materials-13-03462-f011:**
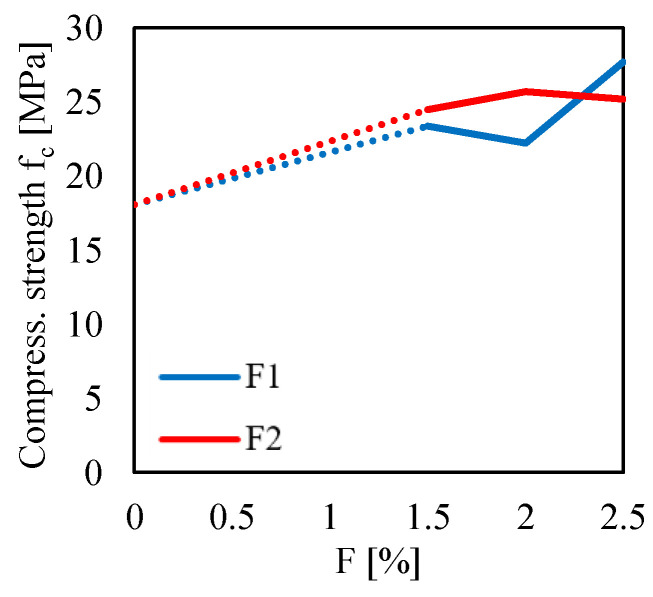
Compressive strength $f_{c}$ measured in the CTs for two fiber type (F1 and F2) with different fiber content.

**Figure 12 materials-13-03462-f012:**
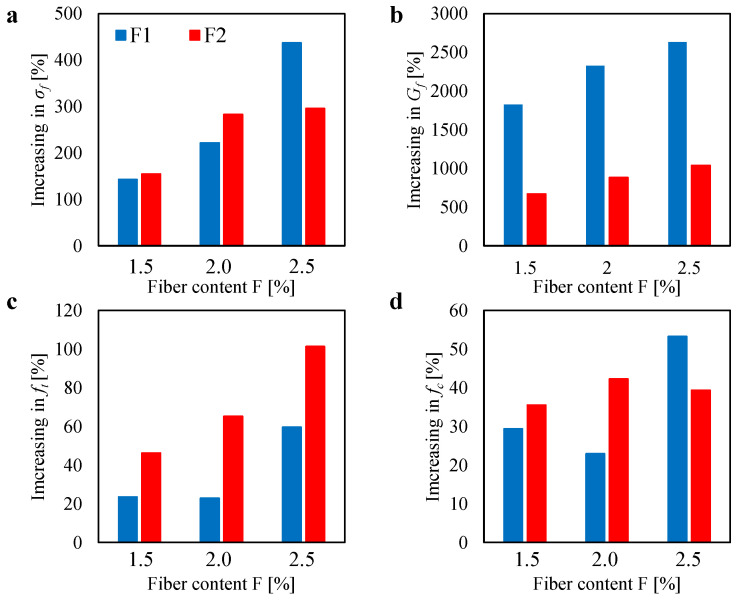
Increase in the mechanical properties (flexural strength $\sigma_{f}$, fracture energy $G_{f}$, tensile strength $f_{t}$ and compressive strength $f_{c}$) with respect to the unreinforced case by varying the fiber content. Figure also shows the comparison between the two fiber types (the F1 and the F2).

**Table 1 materials-13-03462-t001:** Geometrical and mechanical properties of the glass fiber used in the experiments.

Name	$\ell_{f}$ [mm]	$d_{f}$ [mm]	$d_{f}^{*}$ [mm]	$\rho_{fib}$ [kg/m^3^]	$E_{f}$ [MPa]	$f_{t,f}$ [MPa]	$\varepsilon_{u,f}$ [%]	MC [%]
F1	24	0.0140	0.476	2680	72,000	1700	3.7	0.6
F2	13	0.0135	0.316	2680	72,000	1700	3.7	0.5

**Table 2 materials-13-03462-t002:** Mechanical properties of the lime-based mortar reinforced by two types of short-fibers (F1 and F2) with different fiber content *F*.

Name	*F* [%]	$\sigma_{f}$ [MPa]	$G_{f}$ [N/mm]	$f_{t}$ [MPa]	$f_{c}$ [MPa]
NF	0	2.9 ± 0.71	0.32 ± 0.3	2.0 ± 0.1	18.1 ± 2.4
F1	1.5	7.0 ± 0.9	5.0 ± 1.0	2.5 ± 0.3	23.4 ± 0.5
F1	2.0	9.2 ± 0.9	6.3 ± 1.0	2.5 ± 0.3	22.2 ± 2.4
F1	2.5	15.4 ± 0.9	7.1 ± 0.8	3.2 ± 0.7	27.7 ± 4.5
F2	1.5	7.3 ± 1.5	1.9 ± 0.6	2.9 ± 0.5	24.5 ± 3.5
F2	2.0	11.0 ± 1.1	2.7 ± 0.4	3.3 ± 0.2	25.7 ± 6.4
F2	2.5	11.4 ± 0.8	2.9 ± 0.3	4.0 ± 0.3	25.2 ± 1.9

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
