# Peer review of "Lime-Based Mortar Reinforced by Randomly Oriented Short Fibers for the Retrofitting of the Historical Masonry Structure"

_materials, 2020, doi:10.3390/ma13163462_

Round 1

Reviewer 1 Report

114-130 Repetition of sentences or words!

131-139 references

144-149 references

150-158 fiber distribution in the cement

Figure 1 is very poor

Most critical aspects found in the paper are related to lack of demonstration for the proposed conclusions. Examples:

377 absorption level

408 higher bond behavior

437/438 higher distribution of the fiber

Visual techniques should be included to show fiber/cement bond-interface characteristics and fiber distribution (e.g locally can be oriented, due their big sizes)

Reviewer 2 Report

This paper provide very little scientific soundness. Nearly all the theories are easily found in some textbook concerning cementitious materials. Authors cannot point out the significance of the research study. Therefore, this paper should be rejected from my opinion.

Reviewer 3 Report

Some of the English is a little irregular. 

Lines 128-130 repeat lines 114-117

451-452 The statement "Indeed, the higher fiber content would lead to higher homogeneity" may not be true as workability reduces with fibre content. 

Reviewer 4 Report

This is a very interested paper on a research project about the mechanical properties and behavior of composite mortals reinforced by randomly oriented short fibers. The results of this analysis obtained from an extensive laboratory test program which is illustrated and presented very well in the paper. Th results highlight that this composite material ensures excellent ductility capacity and it can be considered a promising alternative reinforcing system of existing masonry structures.

Comment:

  1. The interesting comment including in the sentence “Results plotted in Figure 5 indicate that the F1 fiber leads to higher workability of the product (higher slump value), as compared to the F2 one.” at the end of page 10, needs more comments and discussion in respect to the higher lengths of fibers F1 (24mm instead of 13mm  of F2).
  2. Also the limit to the lengths of the fibers is a very interesting question on this research and this could be mentioned in the text.

Reviewer 5 Report

A very interesting paper.

However, I recommend the authors to justify and explain the use of mortar with only a lime content equal to 30% of the total weight of the binder. Why do you use a cement-lime mortar and not only a lime mortar?

And I suggest them to prove with lime mortars without cement or to use other traditional materials mixed, such as gypsum, in the next research.

Finally, please check the word "piers", maybe it would be better "pillars".

Reviewer 6 Report

Specific comments
row 45
“On the second hand, the use of pultruded rods, which consists of placing
them into grooves cut onto the surface of the member being strengthened.” >> there is no predicate in this sentence. Must be corrected.
row 89
“A common disadvantage between the FRCMs and FRPs…” >> between? rather “of both” or some other expression of an adequate meaning.
row 94
“Fibers activate their characteristics along with their axial direction…” >> delete “with”
row 109,
“First, the samples were tested in three-point bending configuration. Then, the end pieces of the broken beams with a size of 80×40×40 were tested in compression and Brazilian configurations. A final comparison between all the mechanical properties is proposed and analyzed.” >> Such an experimental procedure is an element of Compact Diagnostic Test published in 2006 when a single micro-core is used to measure flexural, compressional and indirect tensile strength of a specimen material. The citation of methodology i.e.
http://reports.ippt.pan.pl/IFTR_Reports_3_2006.pdf or
https://doi.org/10.4028/www.scientific.net/AMR.133-134.223 or another one bibliographical source known to you and presenting such a methodology is suggested.
row 138 and 150
“arranged astride the lesion” >> lesion is a medical term, “damage seems to be more appropriate.
row 102
“However, the fibers were originally treated with dressing by manufacturers.” >> Some comments about dressing type are needed as the dressing influences adherence properties.
row 236
Adequate International Standard is preferred.
rows 279-280
“by dividing the total applied energy with…” >> by dividing the total applied energy by
rows 345-346
“he tests were performed on 40 mm × 40 mm × 80 mm prismatic mortar specimens
obtained from the two-half specimens tested in 3PBT (Figures 4).” >> the two-half length specimens were not tested in 3PBT. They result from 3PBT after testing the full length specimen.
Equations (4) and (5)
Figure 4 comes after the equations so notation used in (4) and (5) is unclear for a reader. One has to guess that “d” means the specimen length in this case. Please make it better readable by explaining the notation.

Round 2

Reviewer 2 Report

Authors put their arguments without any supportive evidence and their reply shows the insufficiency in their research study.